# Multiple Aspects of Inappropriate Action of Renin–Angiotensin, Vasopressin, and Oxytocin Systems in Neuropsychiatric and Neurodegenerative Diseases

**DOI:** 10.3390/jcm11040908

**Published:** 2022-02-09

**Authors:** Ewa Szczepanska-Sadowska, Agnieszka Wsol, Agnieszka Cudnoch-Jedrzejewska, Katarzyna Czarzasta, Tymoteusz Żera

**Affiliations:** Department of Experimental and Clinical Physiology, Laboratory of Centre for Preclinical Research, Medical University of Warsaw, 02-097 Warsaw, Poland; awsol@wum.edu.pl (A.W.); acudnoch@wum.edu.pl (A.C.-J.); kczarzasta@wum.edu.pl (K.C.); tzera@wum.edu.pl (T.Ż.)

**Keywords:** cardiovascular disorders, neuropsychiatric/neurodegenerative disorders, cognition, emotions, stress, COVID-19, analogues of angiotensins, vasopressin, oxytocin

## Abstract

The cardiovascular system and the central nervous system (CNS) closely cooperate in the regulation of primary vital functions. The autonomic nervous system and several compounds known as cardiovascular factors, especially those targeting the renin–angiotensin system (RAS), the vasopressin system (VPS), and the oxytocin system (OTS), are also efficient modulators of several other processes in the CNS. The components of the RAS, VPS, and OTS, regulating pain, emotions, learning, memory, and other cognitive processes, are present in the neurons, glial cells, and blood vessels of the CNS. Increasing evidence shows that the combined function of the RAS, VPS, and OTS is altered in neuropsychiatric/neurodegenerative diseases, and in particular in patients with depression, Alzheimer’s disease, Parkinson’s disease, autism, and schizophrenia. The altered function of the RAS may also contribute to CNS disorders in COVID-19. In this review, we present evidence that there are multiple causes for altered combined function of the RAS, VPS, and OTS in psychiatric and neurodegenerative disorders, such as genetic predispositions and the engagement of the RAS, VAS, and OTS in the processes underlying emotions, memory, and cognition. The neuroactive pharmaceuticals interfering with the synthesis or the action of angiotensins, vasopressin, and oxytocin can improve or worsen the effectiveness of treatment for neuropsychiatric/neurodegenerative diseases. Better knowledge of the multiple actions of the RAS, VPS, and OTS may facilitate programming the most efficient treatment for patients suffering from the comorbidity of neuropsychiatric/neurodegenerative and cardiovascular diseases.

## 1. Introduction

Cardiovascular diseases (CVDs), which are the leading causes of disability-adjusted life years (DALYs), morbidity, and premature death, are particularly challenging when they are associated with diseases affecting the central nervous system (CNS) [1,2,3]. Several studies point to the close cooperation of the CNS and the cardiovascular system (CVS) in the regulation of basic vital functions (Figure 1). The CNS requires a continuous supply of oxygen and nutrients, and the removal of metabolites by the circulating blood, whereas the blood flow needs to be precisely controlled by the autonomic nervous system and the cardiovascular brain regions of the CNS, located at several levels of the brain, including the cortex [4,5,6]. Neurons and glial cells release potent cardiovascular factors which regulate the function of neighboring cells and remote organs, while the heart, vessels, and kidneys, produce neuroactive factors, which are transported to the brain and modulate the function of the CNS [6,7,8,9,10,11]. Substantial evidence indicates that all classical neurotransmitters and several neuroactive peptides, which transmit information between the cardiovascular regions of the brain and the spinal cord, are also engaged in the regulation of cognition, emotions, pain, and behavior at the molecular level [12,13,14,15,16,17]. Moreover, it has been shown that brain diseases are often associated with cardiovascular disturbances [15,18,19,20,21].

Several neuropeptides, initially known as the cardiovascular agents, such as angiotensins, vasopressin, and oxytocin, have been identified as effective modulators of cognitive functions, pain, stress, and emotions [16,17,22,23,24]. Moreover, these compounds frequently interact with other neuroactive agents and their interaction can be altered in cardiovascular pathologies [25,26,27]. It has been found that the effectiveness of second-generation antipsychotics and antidepressive compounds may differ in patients with CVDs, but the reasons for these differences have not been satisfactorily determined [18,28].

The purpose of the present review is to draw attention to the role of cardiovascular neuropeptides in neuropsychiatric and neurodegenerative disorders. We focused on neuropeptides forming the renin–angiotensin (RAS), vasopressin (VPS), and oxytocin (OTS) systems for several reasons. First, these peptides are essential multifunctional molecules when it comes to tuning up the excitatory and inhibitory processes in the brain. Second, the processes of synthesis, release, and the action of these peptides in the CVS and CNS are relatively well recognized. Third, in many instances, the RAS, VPS, and OTS are activated jointly. Fourth, the release of angiotensins, vasopressin, and oxytocin, and the action of these peptides, are altered in CVDs, neuropsychiatric and neurodegenerative diseases. Fifth, some of the specific agonists and antagonists of the RAS, VPS, and OTS are commercially available and used for therapeutic purposes.

## 2. Renin–Angiotensin System

### 2.1. A Brief Overview of the RAS

The main components forming the RAS are shown in Figure 2. Circulating renin is mainly synthesized in the juxtaglomerular cells of the renal afferent arterioles. Renin is necessary to detach angiotensin I (Ang I) from angiotensinogen. Subsequently, Ang I is converted either to Ang II, a potent vasoconstrictive octapeptide, by angiotensin converting enzyme 1 (ACE), or to a vasodilatory peptide angiotensin-(1-7) [Ang-(1-7)] by angiotensin converting enzyme 2 (ACE2). An alternative pathway for the formation of Ang-(1-7) involves the cleavage of Ang-(1-9) from Ang I by ACE2 and the subsequent detachment of Ang-(1-7) by ACE. Ang II can also be metabolized to Ang III and then to Ang IV [29,30,31,32]. ACE is expressed mainly by the endothelial cells of the pulmonary vessels, but it is also present in the brain, heart, and other organs. In addition to the classical hormonal (tissue-to-tissue) RAS, the local paracrine/autocrine (cell-to-cell) RAS and the intracrine (intracellular/nuclear) RAS have both been identified in various organs, including the brain and the heart. The local RAS systems are activated by topical stimuli [11,29,32].

The actions of Ang II and Ang III are mediated by the AT1 receptors (AT1R) and the AT2 receptors (AT2R). Stimulation of AT1R causes the activation of the nicotinamide adenine dinucleotide phosphate (NADPH)-oxidase complex and may promote inflammatory processes [33,34]. Stimulation of AT2R is associated with the activation of phosphotyrosine phosphatases, especially serine/threonine phosphatase 2A, protein kinase phosphatase, and SHP-1 tyrosine phosphatase. It also causes the inactivation of mitogen activated protein kinases (MAPK), specifically p42 and p44 MAPK [35]. The most prominent expression of AT2R has been found in the kidney, heart, blood vessels, and brain, especially in the soma and dendrites of the paraventricular nucleus (PVN) [36]. AT1Rs and AT2Rs may form heteromers, and the blockade of one of the components of this heterodimer increases the interaction of Ang II with the other component [37]. Ang IV stimulates the AT4 receptors (AT4Rs), identified as insulin-regulated aminopeptidase (IRAP) [38], but, in high concentrations, it can also stimulate AT1R. It has been shown that the Ang IV/AT4R pathway in the brain interacts with the hepatocyte growth factor/c-Met receptor system [39]. Ang-(1-7) activates the Mas receptors (MasRs) and the ACE2/Ang-(1-7)/Mas axis [40,41,42].

### 2.2. Systemic RAS and Brain RAS

Active elements of the RAS have been found in the kidney, lungs, adrenal glands, heart, vessels, carotid glomeruli, brain, and the spinal cord [16,31,43,44,45]. Among the stimulators of the RAS are hypoxia, hypovolemia, hypotension, sympathetic stimulation, stress, pain, and specific neuroendocrine factors [7,16,17,31,43,44,46,47,48,49]. The RAS is activated by inflammatory processes associated with tissue injury and oxidative stress. On the other hand, the components of the RAS intensify the inflammatory processes through cooperation with proinflammatory cytokines [10,24,27,50,51,52,53,54]. Acting via the AT1R, Ang II stimulates the NADPH-oxidase complex and NADPH-dependent oxidases that generate inflammatory processes involved in tissue degeneration [34,55].

Prorenin, renin angiotensinogen, angiotensin converting enzyme (ACE), angiotensins I-IV, ACE2, Ang-(1-7) and their receptors (AT1R, AT2R, and MasR) are present in multiple brain regions regulating blood flow, the water-electrolyte balance, and cognitive and emotional processes [31,44,56,57,58].

In the forebrain, components of the RAS have been found in the cortex, the hypothalamus, and the circumventricular organs (organum vasculosum laminae terminalis, OVLT, and the subfornical organ, SFO) [36,59,60,61,62]. They have been detected in the midbrain and hindbrain, specifically in the periaqueductal gray (PAG), the substantia nigra (SN), the dorsal raphe nucleus (DRN), the rostral ventrolateral medulla (RVLM), the caudal ventrolateral medulla (CVLM), the nucleus of the solitary tract (NTS), the nucleus ambiguous (NcAmb), the dorsal motor nucleus of the vagus (DMVNc), and the area postrema (AP) [40,63,64,65,66,67,68,69,70]. MasR mRNA and its protein have been identified in the brain stem cardiovascular regions encompassing the dopaminergic neurons of the substantia nigra [40,71].

In the brain, the central components of the RAS mediate sympathetic stimulation, which is particularly intensive during hypoxia [72,73]. In addition, they regulate the release of several factors involved in the generation of the final cardiovascular response. For example, the local application of prorenin into the supraoptic nucleus (SON) increases membrane excitability and the firing responses of magnocellular neurons, in addition to elevating plasma vasopressin (AVP), with all of these effects being mediated by the inhibition of A-type potassium channels [74]. RAS components have been found in the cranial and spinal cord ganglia. In the spinal cord, AT1R and AT2R have been detected in the regions involved in the regulation of pain and sympathetic outflow (intermediolateral cell column, lamina X and V) [17,56,75].

### 2.3. RAS in Cardiovascular Disorders

The cardiovascular actions of Ang II are exerted by the stimulation of the central and peripheral angiotensin receptors and are partly mediated by the stimulation of neurons regulating the autonomic nervous system or the secretion of cardiovascular hormones, cytokines, and other vasoactive factors acting in the brain or on the systemic circulation [24,27,52,63,64,76,77,78,79,80,81,82,83,84]. Experimental studies have shown that the administration of Ang II directly into the brain exerts a pressor action, which is significantly potentiated in hypertension and cardiac failure, and which is mediated by AT1Rs [24,46,85,86,87]. It is essential to note that Ang II upregulates its own receptors in the brain, specifically in the hypothalamus, the SFO, and the RVLM, and it is likely that, under chronic conditions, it can potentiate its own effects [63,88,89,90,91]. The central administration of AT1R antagonists significantly reduces the cardiovascular effects of the sympathetic stimulation in heart failure and suppresses the activation of the hypothalamic–pituitary axis during isolation stress [22,83]. In the mouse model, the deletion of the *AT1R* gene in the paraventricular nucleus (PVN) has been shown to significantly reduce anxiety-like behavior, blood pressure elevation, heart rate variability, and, in addition, decreases the expression of proinflammatory cytokines in the hypothalamus during exposure to the elevated plus maze test [92]. In contrast, transgenic mice expressing the AT1R in the C1 neurons of the ventrolateral medulla manifested an exaggerated pressor response to aversive cage-switch stress [93]. There is evidence that the central pressor action of Ang II is potentiated by cytokines, and that AT1Rs are involved in the pressor action of interleukin 1 beta (IL-1β) and tumor necrosis factor α (TNF-α) [27,52,55,84]. In this context, it is worth noting that brain inflammation, induced by the systemic administration of lipopolysaccharide, elicits a significant release of the proinflammatory cytokines TNF-α, IL-1β, and interleukin 6 (IL-6) to the systemic circulation and increases the expression of TNF-α, IL-1β, and IL-6 mRNAs in the prefrontal cortex, the PVN, the SFO, the amygdala, and the hippocampus. The above effects can be significantly reduced by the systemic administration of centrally acting AT1R antagonist candesartan [94].

In many respects, the activation of the ACE2 → Ang (1-7) → MasR axis plays the opposite role in blood pressure regulation. An overexpression of ACE2 or the intracerebroventricular administration of Ang-(1-7) reduces the blood pressure in experimental models of hypertension and heart failure [56,70,95,96,97].

### 2.4. Inappropriate Function of RAS in Neuropsychiatric/Neurodegenerative Diseases

The inappropriate functional actions of renin–angiotensin system in selected neuropsychiatric and neurodegenerative disorders are summarized in Table 1.

#### 2.4.1. RAS in Cognitive Disorders

Both experimental and clinical studies have shown that the excessive activation of Ang II receptors may impair cognitive processes [98,99], whereas the blockade of AT1 receptors by a specific AT1R antagonist exerts a neuroprotective action and improves cognitive functions [100,101,102]. In the caucasian population, exposure to ACE inhibitors protected carriers of the AA genotype of the GAG and the CC genotype of the M235T from mental decline [121]. In contrast, Ang IV and Ang (1-7) appear to exert positive effects on cognition and both of these peptides improve learning and memory capabilities [103,104,105,229]. In the brain, Ang-(1-7), Ang IV, and their receptors are present mainly in the neocortex, hippocampus, amygdala, and basal ganglia, and it is likely that these regions may be the main sites of their positive role in cognition [230,231].

#### 2.4.2. RAS in Stress and Pain

Stress provoked by tissue injury, ischemia, hypoxia, inflammation, stroke, or myocardial infarction activates the RAS and increases expression of the AT1R in the brain, heart, and kidney [16,23,25,66,67,85,87,115]. Twenty-four hours isolation stress increases AT1R binding in the PVN and the adrenal medulla and these effects can be abolished by the central or systemic administration of the AT1R antagonist candesartan [232,233]. Experiments on animal models of chronic mild stress have provided evidence for the significant increase of AT1R mRNA expression in the septal/accumbal, diencephalic, medullary, and cerebellar regions of the brain, and in the renal medulla [67]. Furthermore, studies on rats have shown that the cardiovascular responses to chronic mild stress and acute restraint stress are modulated by the RAS and the stimulation of AT1 receptors [86,234]. The effects of microinjections of ACE inhibitors and AT1R and AT2R antagonists directly into the prelimbic cortex give rise to the assumption that Ang II enhances the pressor response to stress by means of the AT1R, while the tachycardic response to stress is enhanced by AT2R [116].

Pain, which is one of the stress-inducers and which is frequently experienced in cardiovascular and neuropsychiatric/neurodegenerative diseases, is a potent activator of the RAS. The specific components of the RAS play a complex role in the regulation of pain [17,235,236]. In the spinal cord, heart, and other organs, Ang II induces nociceptive action, whereas both stimulation of the AT1R and AT2R and the activation of the Ang-(1-7) → MasR pathway in the brain elicit analgesia [117,118].

#### 2.4.3. RAS in Depression and Anxiety

The clinical picture of depression manifests significant heterogeneity. Inappropriate dimensions, the activity and metabolism of the cortex, the subcortical limbic regions, the basal ganglia, and the brain stem may suggest that the pathogenesis of depression is associated with the improper function of these structures [237,238]. The role of the dorsolateral prefrontal cortex, the anterior cingulate cortex, the orbital frontal cortex, and the insula, appears to be particularly interesting. These regions have multiple connections with the amygdala, the thalamus, the lateral and medial orbitofrontal cortex, and the medial prefrontal cortex, which are all involved in the regulation of emotions [239,240,241]. Magnetic resonance imaging studies performed in patients with depression revealed reduced dimensions of the frontal and orbitofrontal lobes, and decreased metabolism and blood flow in the dorsolateral prefrontal cortex and the ventral region of the anterior cingulate cortex [237,242,243]. In addition, it has been found that a deficit in facial disgust recognition correlates with reduced grey matter volume in the insula, which is engaged in the regulation of emotions and blood pressure regulation [244].

Earlier studies could not confirm any associations between ACE I/D genotypes and depression [119,245]; however, the significant association between depression and the AT1R A1166C CC genotype, found by Saab et al. [119], suggests that the inappropriate synthesis of AT1R may contribute to the development of depression in patients. More recently, an analysis of haplotype-tagging single nucleotide polymorphism of angiotensin AT1R revealed significant differences between the cohorts of depressed and nondepressed patients with rs10935724 and rs12721331 htSNPs. The authors found significant associations between AT1R htSNPs and the volumes of the prefrontal cortex and the hippocampus [246].

Experimental studies on transgenic rats [TGR(ASrAOGEN)680] have shown that rats with low brain angiotensinogen manifest anxiety-related behavior and depressive-like behavior, which can be reversed by the ICV application of Ang-(1-7) [120]. The authors suggest that anxiety and depression in these rats may be caused by a deficit of Ang-(1-7), which is one of the derivatives of angiotesinogen (Figure 2).

#### 2.4.4. RAS in Alzheimer’s Disease

The extracellular deposition of aggregated amyloid beta (Aβ) plaques and the formation of neurofibrillary tangles of hyperphosphorylated tau protein as well as the suppressed function of the brain cholinergic system are characteristic features of Alzheimer’s disease (AD), but the mechanism of Aβ toxicity is not yet fully understood [247,248]. Elevated concentrations of ACE, angiotensin II, and AT1 receptors in the cerebral cortex of patients with AD suggest that enhanced activation of the RAS may inhibit the release of acetylcholine in the cortex and contribute to the development of Alzheimer’s dementia [106,107,108]. Losartan and valsartan decrease Aβ peptide oligomerization in primary neuronal cultures and reduce cognitive impairment in Tg2576 AD transgenic mice, expressing the human 695-aa isoform of the amyloid precursor protein gene (APP) [249]. In addition, captopril prevents Aβ-induced downregulation of some genes involved in neuronal regeneration and cognition [250]. Studies performed on animal models of AD and postmortem examinations of human brains has led investigators to suggest that the excessive activation of the brain AT1R and insufficient activation of AT2R may induce excessive generation of reactive oxygen species (ROS), and this may account for the prevalence of neurodegenerative processes over the neuroprotective processes in the brains of AD patients [109,110,111,112,113].

Studies exploring the effects of ACE inhibitors and AT1R blockers on the cognitive abilities of AD patients have not yielded uniform results. Symptoms of dementia tend to be lower in patients treated with ACE inhibitors; however, the differences are not significant [98]. Similarly, a lack of the beneficial effects of RAS targeting compounds on dementia and AD symptoms was reported in the ONTARGET and TRANSCEND clinical trials [251], and in a quantitative meta-analysis [252]. However, in another study, a significantly slower progression of AD was found in patients treated with ACE inhibitors crossing the blood–brain barrier (captopril, perindopril) [253]. More recently, the memory improving effects of ACE inhibitors and AT1R blockers were found in a meta-analysis of patients with AD and cognitive impairment of aging [254,255], as well as in a cohort study in which AT1R blockers were applied together with statins [256].

It has been found that the Ang-(1-7) level is significantly reduced in the cerebral cortex and hippocampus of the senescence-accelerated mouse prone 8 (SAMP8) model of Alzheimer’s disease and that the inhibition is associated with an inverse correlation between the Ang-(1-7) level and tau hyperphosphorylation. Therefore, it has been suggested that the inappropriate activation of the Ang-(1-7)/Mas axis may play a role in the pathogenesis of AD [114].

#### 2.4.5. RAS in Parkinson’s Disease and Tardive Dyskinesia

Cardiac autonomic dysfunction, orthostatic hypotension, and ECG abnormalities belong to the most common non-motor symptoms of Parkinson’s disease (PD) and may even precede motoric disorders [19,257]. Parkinson’s disease is associated with a loss of dopaminergic neurons in the pars compacta of the substantia nigra. With the advancement of the disease, progressive degeneration and dysfunction occurs in other classical and nonclassical neurotransmitter systems, including the RAS [258]. The accumulation of α-synuclein, which has been proposed to be one of the cytotoxic PD factors inducing microglial activation, is associated with the increased expression of AT1Rs and NADPH oxidase activation. It has been shown that the blockade of AT1Rs by candesartan and telmisartan significantly reduces the negative effects of α-synuclein in microglia and dopaminergic neurons [51]. In the rat PD model produced by the administration of 1-methyl-4-phenyl-1,2,3,6-tetrahydropyridine (MPTP), the application of AT1R antagonists or ACE inhibitors significantly reduces neuronal cell death. Similar effects were obtained in primary mesencephalic cell cultures. It has been suggested that endogenous Ang II potentiates the neurotoxic effect of MPTP on dopaminergic neurons, and that the antagonists of ACE or AT1Rs exert their beneficial effects through the inhibition of microglial and NADPH activation and the suppression of the prooxidative and proinflammatory effects mediated by cytokines [125,126,127]. Later studies have shown that the administration of candesartan does not influence motor efficiency nor the dopamine and serotonin levels in the striatum, but it increases the expression of dopamine D1 receptors and decreases the expression of dopamine D2 receptors. Studies on the rat model of Parkinson’s disease have indicated that the neuroprotective effect of chronic treatment with AT1R antagonists is associated with the formation of heterodimers of AT1R/AT2R [37,125]. It is worth noting that the administration of candesartan and lisinopril reduces the release of proinflammatory cytokines (IL-1β, TNF-α) and glutamate in the rat model of haloperidol-induced tardive dyskinesia, which is another dopaminergic disorder resulting from damage of the striatal neurons [128].

To date, meta-analyses on human subjects have not provided explicit evidence for the association between ACE gene I/D polymorphism and PD risk [259].

#### 2.4.6. RAS in Schizophrenia and Autism

Schizophrenia belongs to difficult neuropsychiatric disorders which affect human beings of different ages and sexes. Unfortunatelly, relatively little attention has been given to the regulation of the cardiovascular system in schizophrenic patients.

In comparison with the general population, patients with schizophrenia are more predisposed to CVDs and have a greater risk of adverse cardiac events, such as stroke and heart failure [260,261,262,263,264]. On the other hand, patients with schizophrenia and myocardial infarction who receive secondary preventive cardiovascular treatment have a similar mortality rate as the general population [265].

It is only in a few studies that attempts have been made to find out whether there are relations between mutations of RAS genes and schizophrenia, and the results are not uniform. Studies on cardiovascular genomics and cognitive function in patients with schizophrenia revealed some missense mutations of angiotensinogen (AGTM268T, AGT235T). The mutations involved replacement of valine by threonine and were associated with the decline of cognitive functions and lower verbal memory scores [121,122]. An association between ACE I/D (insertion/deletion) polymorphism and a disposition to schizophrenia was found in the study of Gadelha et al. [123], but not in the earlier report of Gard et al. [245], nor in the meta-analysis collecting data from European, Asian, and Turkish populations [266]. The alleviation of schizophrenia symptoms in patients with the disorder via the AT1R antagonist telmisartan was found by Fan et al. [124]. The beneficial effects of AT1 receptor blockade with irbesartan in schizophrenic patients with psychogenic polidypsia may suggest that the inappropriate activation of some of the components of RAS may account for polidypsia in patients with schizophrenia [267].

Strong associations between DD genotype of ACE I/D and the D allele has been found in autistic patients and the authors suggest that genetic diversity of RAS may enhance the risk of autism [268].

#### 2.4.7. RAS in Coronavirus Infections

Growing evidence indicates that the inappropriate activation of RAS components may contribute to disturbances in brain function in coronavirus disease (COVID-19) and that survivors of COVID-19 manifest aggravated symptoms of neuropsychiatric disorders such as cognitive and attention deficits (i.e., brain fog), new-onset anxiety, depression, psychosis, seizures, and suicidal behavior [269]. A substantial body of evidence indicates that COVID-19 infection affects the RAS and that the imbalance of the local and systemic Ang II and Ang-(1-7) activities may play an essential role in the pathological processes developing in the lungs and other organs during SARS-CoV-2 infections [270,271]. Human ACE2 binds the virus S protein and plays a fundamental role in transmitting the original SARS-CoV and the new SARS-CoV-2 to the targeted cells [272]. Expression of ACE2 mRNA and its protein has been reported in the cortex, striatum, hippocampus, brain stem, and cerebrovascular endothelium of the rodent brain [273,274,275] and the human brain [276,277]. SARS-CoV-2 is expressed preferentially in cultured glial cells, specifically in astrocytes and radial glial progenitor cells [278,279]. Preclinical studies on transgenic mice suggest that SARS-CoV, very closely related to the SARS-CoV-2 coronavirus, can access the brain through the olfactory nerve and the olfactory bulb, and it may subsequently be transported transneuronally or spread via the Virchow–Robin spaces and along brain vessels causing extensive neuronal infection [280]. The SARS-CoV-2 spike protein, which readily crosses the blood–brain barrier, induces an inflammatory response within microvascular endothelial cells, leading to the dysfunction of the blood-brain barrier [276]. In the brains of infected patients, SARS-CoV has been detected almost exclusively in neurons [281,282]. The neuroinvasive capacity of the SARS-CoV-2 virus has also been reported. For example, autopsies of brains from patients who died of COVID-19 made it possible to detect SARS-CoV-2 in the cortical neurons [283].

It is likely that the inappropriate function of the RAS may contribute to the exaggeration of psychiatric symptoms in patients with COVID-19, especially when considering that the excessive stimulation of the AT1R influences microglial polarization and induces an active M2a proinflammatory state, thereby initiating neurodegenerative processes [42]. This assumption is highly feasible in view of the data showing a correlation between neuroinflammation, brain microvascular injury, and cognitive function impairment [129].

Some patients with COVID-19 infection may exhibit so-called “silent hypoxemia”, which is manifested by severe hypoxemia without dyspnea or tachypnea and suggests inadequate stimulation of the arterial chemoreceptors. Since carotid bodies possess local RAS elements, including ACE2, it has been hypothesized that the ACE2-mediated entry of SARS-CoV-2 into carotid bodies may contribute to the development of silent hypoxemia in COVID-19 infection [284,285,286,287]. This notion is further supported by evidence showing the expression of SARS-CoV-2 in carotid bodies from a patient with COVID-19 [288]. Altogether, it appears that the inappropriate activity of RAS should be taken into account as an effective cause of brain dysfunction occurring in COVID-19 disease.

## 3. Vasopressin and Oxytocin Systems

In many instances, vasopressin and oxytocin are released jointly and engage similar cellular mechanisms, acting either synergistically or antagonistically [289].

### 3.1. Overview of Systemic and Peripheral VPS and OTS

Vasopressin and oxytocin are synthesized mainly in the supraoptic, paraventricular, and suprachiasmatic nuclei of the hypothalamus and released to the blood in the neurohypophysis. They are also synthesized in some other cells of the central nervous system and in the peripheral organs [17,289]. The neuroregulatory and peripheral actions of vasopressin are mediated by three types of AVP receptors: V1a (V1aR), V1b (V1bR), and V2 (V2R) [289,290]. Oxytocin stimulates its own receptors (OXTR); however, in higher concentrations, it can also act by means of V1aR [291].

Current evidence indicates that cardiovascular and neuroregulatory processes are tuned up mainly by the V1 receptors [292,293,294,295,296]. AVP cooperates with the RAS through multiple actions exerted in the brain and in the peripheral organs, and its action is significantly altered in cardiovascular diseases [26,85,140,141,292,297,298,299]. The inappropriate functions of the VPS and OTS have been described in neuroregulatory disorders, and it has been suggested that they may play essential roles in the development of social, emotional, and cognitive dysfunctions, including dementia [300,301,302,303,304,305,306,307]. It has been shown that neurogenic stress provokes the release of AVP and oxytocin (OT) and that these two peptides are involved in the regulation of the cardiovascular, emotional, and behavioral responses, and in processing social information [26,140,141,142,143,144,145]. Studies on rodents indicate that the AVP → V1bR pathway plays an important role in the regulation of the hormonal and behavioral responses to stress and in the formation of social recognition memory. Experiments on *Avpr1b*-/-mice suggest that social memory is regulated by the V1bR located in the CA2 region of the hippocampus [154,300] and that the blockade of V1bR induces anxiolytic actions in various models of depression [146,147].

### 3.2. Vasopressin and Oxytocin in Depression

The inappropriate functional actions of VPS and OTS in selected neuropsychiatric and neurodegenerative disorders are summarized in Table 1.

#### 3.2.1. Vasopressin and Depression

Post-mortem examinations of human brains revealed the elevated expression of AVP mRNA in the PVN and SON and a significantly greater number of AVP and OT neurons in the PVN of patients with MDD (major depressive disorder) [130]. In rodent models of anxiety and depression, the application of orally active V1bR antagonists reduced hyperemotionality and elicited anxiolytic- and antidepressant-like effects [137,138,139]. Furthermore, oral administration of a V1bR antagonist (TASPO390325) antagonized the elevation of adrenocorticotropic hormone levels induced by the joint application of corticotrophin-releasing hormone and vasopressin analog (desmopressin, dDAVP) [139]. In the rat model of depression, the blockade of the central V1 receptors abolished anhedonia elicited by chronic mild stress [140].

Indirect evidence for the engagement of the VPS in the pathogenesis of MDD and aggression comes from studies of the V1bR gene polymorphism [131,132]. The disposition to affective disorders was associated with the polymorphism of the V1bR gene, in particular, with the presence of the haplotype associated with A-T-C-A-G for the single nucleotide polymorphism (SNP) s1-s2-s3-s4-s5 allele. It appears that presence of this haplotype may protect from recurrent MDD [131]. A linkage between the V1bR genetic variation SNP rs33990840 and a predisposition to suicidal behavior was also reported [133], and clinical trials assessing the usefulness of V1bR antagonists for the treatment of MDD have recently started [134,135].

Earlier studies assessing the usefulness of blood AVP measurements in patients with depression or mania did not yield uniform results [308,309,310,311,312], and the meta-analyses did not find support to diagnose depression based on the estimations of AVP and OT levels in blood, saliva, urine, and the cerebrospinal fluid (CSF) [306,313]. However, measurements of copeptin, which is a surrogate marker of AVP, indicate that blood copeptin levels are elevated in patients who are insensitive to antidepressant treatment, and it appears that copeptin may be a useful biomarker for the early selection of non-responders to specific antidepressant treatments [136].

#### 3.2.2. Oxytocin in Depression and Anxiety

A prevailing number of studies indicate a correlation between the altered function of the central oxytocinergic system and affective disorders, although the results of human studies are inconsistent. Most of the studies report lower levels of plasma OT in patients with MDD and bipolar affective disorder during depressive episodes than in the control subjects [184,187,188,189]; however, higher [190,191] or normal [192] levels were also found. It appears that plasma OT levels may be lower in MDD than in the control subjects [187,314] for women in particular, which may suggest the greater sensibility of the female OT system to disturbed signaling in affective disorders. Similarly, measurements of the OT level in the CSF did not give explicit results. Demitrack and Gold did not find significant differences in OT levels in the CSF between MDD patients and the control group [315]. Analogous results were obtained in other studies [316,317]. On the contrary, postmortem studies to assess the neuronal OT mRNA levels in MDD patients have indicated an elevated activation of the OT system in MDD patients. Namely, using immunocytochemical techniques on the post-mortem samples of the PVN collected from patients with MDD and bipolar affective depressive episode, Purba et al. [130] found significant elevation of OT-immunoreactivity in patients with depressive mood disorders. These results were supported by a case-control study, in which elevated OT-immunoreactivity in the PVN was found in MDD and bipolar disorder patients [185]. The increased expression of OT mRNA in the PVN was also reported in melancholic MDD patients [186]. Recently, post-mortem estimations of OT receptor (OTR) mRNA expression have shown that patients with MDD and bipolar disorder have significantly higher OTR mRNA levels in the dorsolateral prefrontal cortex, a brain structure implicated in the pathophysiology of numerous psychiatric disorders including MDD and bipolar disease [318]. Altogether, the available data indicate the increased activity of the central OT system in depression. However, it should be emphasized that the intracerebral and peripheral release of OT in depression may occur either in a coordinated or independent manner depending on the quality or the strength of the stimulus [319].

It should also be noted that plasma oxytocin levels positively correlate with help-seeking intentions and behavior in depressed patients [193]. Anderberg and Uvnäs-Moberg found a negative correlation between the plasma OT concentration and the scored symptoms of depression and anxiety [184]. They also reported a positive correlation between the estimation of happiness and the plasma oxytocin level. It has been suggested that plasma OT levels may help to predict whether a patient with chronic depression will respond to psychotherapy [194].

The involvement of OT in mechanisms underlying depression and anxiety has also been analyzed in studies based on genetic tests. In humans, a positive association between depression and separation anxiety was found for a single nucleotide polymorphism (SNP; rs53576) of the OTR gene [195]. Specifically, the GG genotype of this SNP has been linked to high levels of separation anxiety and insecure attachment in MDD patients [196]. Other authors found an interaction between another OTR SNP polymorphism (rs2254298) and the symptoms of depression and anxiety in adolescent girls, especially when the polymorphism occurred in association with an adverse parental environment [197]. More recent data indicate that variations in the OTR and G-protein (Gβ3 rs5443) genes are specifically associated with separation anxiety and depressive symptoms both in childhood and in adulthood [198]. Other authors reported the negative influence of the coincidence of the presence of the OTR rs53576 genotype A allele and environmental adversity in postnatal life (for example, maternal postpartum depression) on the mental health and social behavior of the child [199]. In addition, studies of a SNP of the OTR gene in depressed adults indicate the association of the A allele rs53576 with a history of suicide attempts [200].

Special attention should also be given to the correlation between the dysregulation of the OT system and postpartum depression (PPD). Data from animal models and human studies have suggested that disruptions in the activity of the OT system may account for the relationship between breastfeeding, stress coping, and mood [201,320]. It has been shown that responsiveness to stress is decreased in breastfeeding women (the stimulus for OT secretion) and that this is associated with a stress-induced rise in plasma cortisol [320]. Furthermore, it has been found that lowered plasma OT levels in the third trimester of pregnancy makes it possible to predict postpartum depressive symptoms [321]. In addition, OT levels were inversely correlated with depressive symptoms in mothers who intended to breastfeed in both the third trimester and at 8 weeks postpartum. During breastfeeding, OT release was lower in depressed mothers than in nondepressed mothers [202].

Despite promising results from experimental studies, clinical trials have yielded inconclusive results, with some reports supporting a significant positive effect and others suggesting a null effect in the treatment of anxiety and depression [322,323,324,325]. However, it has been found that oxytocin improves the therapeutic effects of other antidepressants, such as escitalopram [326]. Additionally, intranasally applied OT was shown to modify neural activity in the limbic regions of depressed patients [327]. In a group of Vietnam veterans suffering from post-traumatic stress, the intranasal administration of OT reduced physiological responses during personal combat imagery [328].

#### 3.2.3. Vasopressin and Oxytocin in Alzheimer’s Disease

Patients suffering from Alzheimer’s disease have a lower level of AVP in the CSF [148], and post-mortem studies revealed a reduced expression of AVP immunoreactivity in the hippocampus, nucleus accumbens, and the internal portion of the globus pallidus of AD patients in comparison with controls [149]. Other post-mortem studies of human brains provided evidence for the reduced number of AVP expressing cells in the suprachiasmatic nucleus in senescence and AD patients [150]; however, the vasopressinergic innervation of the PVN, SON, and locus coeruleus in AD patients and non-demented controls did not differ [151,152].

Recently, experiments on the APP/PS1 mouse model of AD have shown that the intranasal application of an AVP derivative [AVP-(4-8)] markedly improves working memory and long-term memory in this experimental model of AD [153].

A number of studies have reported the impact of intranasally applied OT on human cognitive functions (see [178] for a review); however, most of the previous studies did not find significant changes in the activity of the central oxytocinergic system in different brain regions of patients with Alzheimer’s disease [151,179,180,181,182]. A trend for elevated hippocampal OT immunoreactivity in post-mortem brain samples of patients with AD was observed in one of the investigations [180]. More recent studies suggest that the OTS may be affected to some extent in AD patients. Lardenoije et al. found changes in the methylation of the OT gene in patients with AD [329]. Recently, an increase of the OT signal value and plasma OT concentration were correlated with the right parahippocampal gyrus volume in MRI images from the AD group but not in the images from the control group [183].

#### 3.2.4. Vasopressin and Oxytocin in Autism

A growing body of evidence indicates that AVP and OT may play a role in the etiology of autism spectrum disorder (ASD) [172,176,177,330,331]. AVP concentrations in the CSF were found to be lower in children with autism, and AVP levels were associated with the severity of symptoms. Moreover, the AVP concentration in the CSF in neonates made it possible to predict a subsequent diagnosis of autism. No such associations were found for oxytocin; however, both OTR and V1aR mRNA levels were lower in autistic patients [169,170,171]. Studies on gene polymorphism of AVP receptors in autistic patients of the Korean population showed a significant association of ASD with SNP RS1 and SNP RS3 in the 5′flanking regions of the V1aR receptor [173,174,175]. The association between polymorphism of the V1aR and V1bR genes and autism was also studied in the North American population [172]. The authors reported that there may be a significant link between ASD and polymorphism of the AVP *V1bR* in SNP rs35369693 and SNP rs28632197. An improvement of social abilities and a reduction of anxiety symptoms after 4–5 weeks of intranasally applied AVP was reported by Parker et al. [176,177]. A systematic review and meta-analysis of the long-term intranasal application of OT in ASD showed that this type of treatment is well tolerated and safe [228]. The positive effects of AVP on social behavior in ASD have been shown in the studies of Hendaus et al. [332] and Parker et al. [177].

There is evidence for an association between OTR gene polymorphism and susceptibility to ASD. Based on a study encompassing 314 autism-affected families, Ylisaukko-Oja et al. suggested that the p24–26 region of chromosome 3 expressing the OTR gene may play a role in increasing susceptibility to the development of ASD [210]. In the Chinese population, polymorphism was present in the rs2254298 and rs53576 genes [211], in the Caucasian and Japanese populations, it was present in rs2254298 [212,213], in the European population, it was present in rs237887 [214], and in the North America population it was present in rs2268493, rs1042778, and rs7632287 [172,215]. Two meta-analyses, encompassing 8 studies [216] and 10 studies [217], reported associations between ASD and OTR SNP polymorphism in rs2254298, rs7632287, and rs2268491.

Lower plasma oxytocin levels have been observed in children with ASD [218,219,220,221]. Moreover, ASD subjects had lower levels of the bioactive amidated OT form and higher OT precursor levels, suggesting the altered processing of the OT peptide in the brains of children with autism [218]. Several studies have shown significant positive effects of OT application on autistic behavior. OT infusion was shown to ameliorate repetitive behavior in adults with ASD and Asperger’s disorder [222]. This finding was supported by a recent clinical trial based on a group of 106 ASD patients, in which intranasally applied OT reduced repetitive behavior and increased the time of gaze fixation on socially relevant regions [223]. In the same study, however, no improvement was found in the primary outcome, the Autism Diagnostic Observation Scheduled (ADOS)—a social reciprocity subscale with regard to the prevalence of adverse events [223]. Other randomized crossover trials performed on children with ASD showed that treatment with intranasal OT improved caregiver-rated social responsiveness [176,333]. In addition, intranasally applied OT enhanced learning in response to social targets and feedback, and this was correlated with the activation of the nucleus accumbens detected by functional MRI [224]. Other studies failed to find beneficial effects in relation to OT in ASD [225,226,227]. Although the present evidence is promising, further clinical studies are necessary to provide better insight into the role of oxytocin in the pathogenesis of ASD and the potential utility of this peptide in the treatment of ASD in humans.

#### 3.2.5. Vasopressin and Oxytocin in Schizophrenia

Experimental and clinical studies indicate that the dysregulation of the VPS and OTS may play a role in the pathogenesis of schizophrenia [302,303,306,334]. Schizophrenia-like symptoms with impairment of social behavior were described in vasopressin-deficient (di/di) Brattleboro rats [165,166] and *V1aR* knockout mice [167]. In addition, significantly lower concentrations of AVP receptors in the prefrontal cortex and hypothalamus were found in the rat model of schizophrenia induced by prenatal exposure to methylazoxymethanol acetate (MAM) [168].

Post-mortem studies revealed lower AVP levels in the temporal cortex of schizophrenic patients [155]. Similarly, reduced AVP mRNA expression was found in the PVN of schizophrenic patients [156]. The blood AVP levels in patients with schizophrenia were either elevated or not altered [157,158,159]. Interestingly, a positive correlation was found between the blood AVP level and the severity of symptoms in female patients but not in male patients [159]. Some schizophrenic patients with inappropriately high blood AVP levels manifested polydipsia, hypoosmolality, and hyponatremia, which were further potentiated by antipsychotic treatment [160,161,162]. The intranasal application of DDAVP—a synthetic analog of AVP—increased the effectiveness of risperidone in reducing the negative symptoms of schizophrenia [163].

Significant associations between single nucleotide polymorphisms of AVP and OT genes and schizophrenia were detected in the chromosomal region 20p13, specifically in the loci of rs2740204 of the shared promoter of AVP and OT, in rs4813626 of the 5′promoter of OT, and in rs3011589 of the second intron of the AVP promoter [164].

Plasma OT concentrations were found to be lower in patients with schizophrenia than in healthy controls [158,203], and there was a negative correlation between OT levels and symptom severity in patients with schizophrenia [157]. Experiments on rats with the MAM model of schizophrenia revealed reduced concentrations of oxytocin and OTRs in the prefrontal cortex (PFC) and in the hypothalamus of the MAM schizophrenic model [168].

Positive effects related to the intranasal application of OT on social cognition and interpersonal reactivity have been reported in patients with schizophrenia [204,205]; however, clinical trials and meta-analyses have not provided evidence for a significant therapeutic effect of OT in relation to schizophrenia [206,207,208,209].

## 4. Conclusions

A survey of the literature shows that the RAS, VPS, and OTS, the classical endocrine systems regulating blood pressure and the water-electrolyte balance, are also potent regulators of other CNS processes through their actions on cerebral blood flow, the metabolism, and intercellular and intracellular signal transmission. Angiotensinogen, angiotensins, AVP, OT, and their respective receptors have been detected in the neurons, glial cells, and blood vessels of multiple brain regions regulating cardiovascular functions, pain, emotion, susceptibility to stress, as well as learning, memory, and cognitive processes. The RAS, VPS, and OTS innervate the same regions of the brain and in many instances are activated jointly and interact at the cellular level. The components of the RAS, VPS, and OTS have been identified in the forebrain (cortex, hypothalamus, circumventricular organs) and in the midbrain and hindbrain (PAG, DRN, RVLM, CVLM, NTS, NcAmb, DMVNc, AP). It has been shown that the RAS, VPS, and OTS act differently during stress, depression, and anxiety, as well as in neuropsychiatric and neurodegenerative diseases such as Alzheimer’s disease, Parkinson’s disease, autism, and schizophrenia (Table 1). Mutations of angiotensinogen, AT1Rs, and ACE genes were detected in some patients with depression and schizophrenia. In addition, polymorphism of V1bR and OTR genes was demonstrated in patients with depression and autism. It appears that the RAS, VPS, and OTS form a multifunctional cooperating triad which may have a significant impact on the efficacy of therapies of neurodegenerative diseases and psychiatric disorders. Clinical trials and meta-analyses indicate that specific compounds interfering with the action of the RAS, VPS, or OTS may improve the effectiveness of the treatment of neuropsychiatric and neurodegenerative diseases; however, further investigations are needed to establish the guidelines for their use for medical purposes. The present study does not address the putative role of these systems in other essential neurological and psychiatric disorders, such as bipolar disorder, ADHD, epilepsy, migraine headaaches, and additional resaerch in this field should be conducted. Better knowledge of the mechanisms of the actions of these compounds should be helpful in programming the most efficient individually-tailored treatment for patients suffering from the comorbidity of neuropsychiatric/neurodegenerative and cardiovascular diseases.

## Figures and Tables

**Figure 1 jcm-11-00908-f001:**
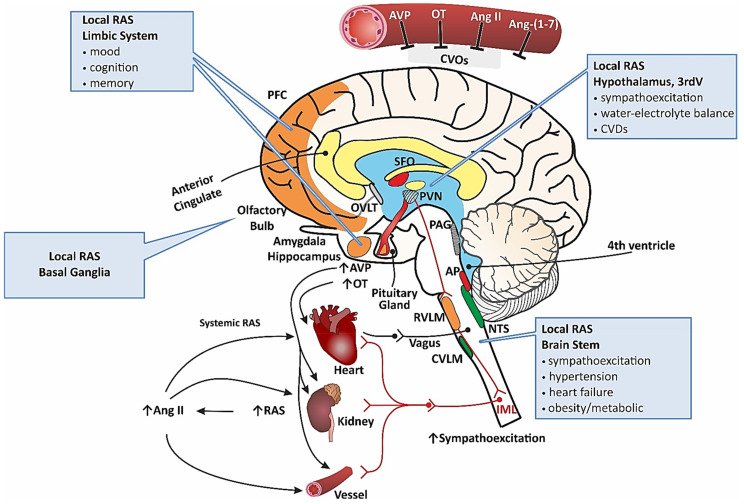
The brain structures involved in the regulation of the cardiovascular, cognitive, emotional, and behavioral functions through actions exerted by the renin–angiotensin system (RAS), and the vasopressin (AVP) and oxytocin (OT) systems. Abbreviations: Ang II—angiotensin II; Ang-(1-7)—angiotensin-(1-7); AP—area postrema; CVLM—caudal ventrolateral medulla, CVOs—circumventricular organs; IML—intermediolateral column; NTS—nucleus of the solitary tract; OVLT—organum vasculosum laminae terminalis; PAG—periaqueductal gray; PFC—prefrontal cortex; PVN—paraventricular nucleus; RVLM—rostral ventrolateral medulla; SFO—subfornical organ; 3rdV—third ventricle.

**Figure 2 jcm-11-00908-f002:**
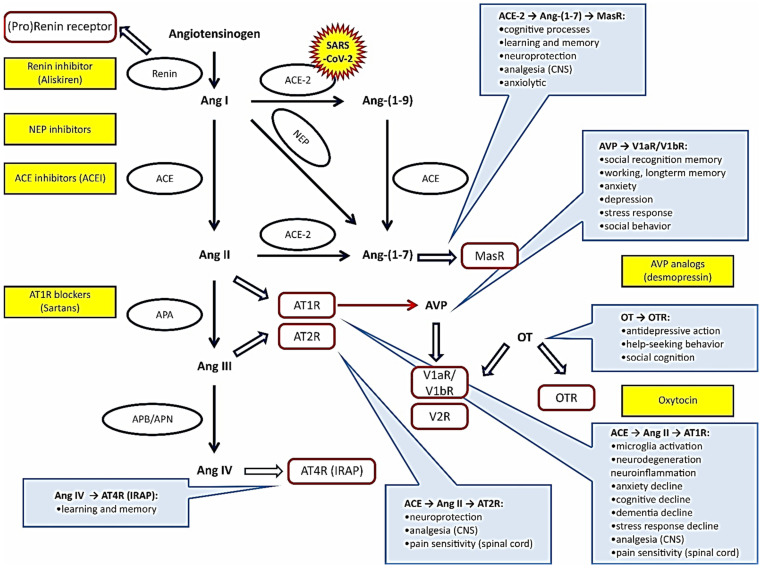
The main components of the renin–angiotensin system (RAS) engaged in the regulation of cardiovascular, cognitive, emotional, and behavioral functions. Abbreviations: ACE—angiotensin converting enzyme; ACEI—inhibitor of ACE; Ang—angiotensin; APA, APB, and APN—aminopeptidases A, B, and N; AT1R, AT2R—angiotensin receptors; AVP—arginine vasopressin; CNS—central nervous system; IRAP—insulin-regulated aminopeptidase; MasR—Mas receptor of Ang-(1-7); NEP—neutral endopeptidase; OT—oxytocin; OTR—oxytocin receptor; V1aR, V1bR, V2R—vasopressin receptors. See also refrences [16,17].

**Table 1 jcm-11-00908-t001:** Summary of the inappropriate functional actions of renin–angiotensin, vasopressin, and oxytocin systems in selected neuropsychiatric and neurodegenerative disorders.

Neuropsychiatric/Neurodegenrative Disorder	Functional Action	References
Renin–angiotensin system (RAS)		
Cognitive disorders	Human and rodent studies:	
	Ang II may impair cognitive processes, probably via AT1R;	[98,99,100,101,102]
	Rodent studies:	
	Ang IV and Ang-(1-7) may improve learning and memory;	[103,104,105]
Alzheimer’s disease	Human studies:	
	The enhanced activation of the RAS may inhibit acetylcholine release in the cortex and contribute to the development of AD dementia;	[106,107,108]
	Human and rodent studies:	
	The excessive activation of the brain AT1R and insufficient activation of AT2R may induce excessive generation of ROS, and this may account for the prevalence of neurodegenerative processes over neuroprotective processes in the brains of AD patients;	[109,110,111,112,113]
	Rodent studies:	
	The inappropriate activation of the Ang-(1-7)/Mas axis may play a role in the pathogenesis of AD;	[114]
Stress and pain	Human and rodent studies:	
	Stress provoked by tissue injury, ischemia, hypoxia, inflammation, stroke, or myocardial infarction, as well as chronic mild stress, activates the RAS and increases expression of AT1R in the brain, heart, and kidney;	[16,23,25,66,67,85,87,115]
	Rodent studies:	
	Ang II enhances the pressor response to stress by AT1R, while the tachycardic response to stress is enhanced by AT2R;	[116]
	The stimulation of AT1R and AT2R, and the activation of the Ang-(1-7) MasR pathway in the brain reduces pain;	[117,118]
Affective disorders	Human studies:	
	A significant association between depression and the AT1R A1166C CC genotype;	[119]
	Ang (1-7) has an antidepressant effect;	[120]
Schizophrenia	Human studies:	
	In patients with schizophrenia, missense mutations of angiotensinogen (AGTM268T, AGT235T) with replacement of valine by threonine are associated with the decline of cognitive functions and lower verbal memory scores;	[121,122]
	An association between ACE I/D (insertion/deletion) polymorphism and a disposition to schizophrenia was found;	[123]
	The AT1R antagonist telmisartan can alleviate the symptoms of schizophrenia;	[124]
Parkinson’s disease	Rodent studies:	
	Increased expression of AT1R and NADPH oxidase activation;	[51]
	Endogenous Ang II potentiates the neurotoxic effect of MPTP on dopaminergic neurons, whereas ACE or AT1Rs antagonists exert their beneficial effects through the inhibition of microglial NADPH activation and the suppression of prooxidative and proinflammatory effects mediated by cytokines;	[125,126,127]
	Chronic treatment with AT1R antagonists is associated with the formation of heterodimers of AT1R/AT2R;	[37,125]
Tardive dyskinesia	Rodent studies:	
	The administration of candesartan and lisinopril reduces the release of proinflammatory cytokines (IL-1β, TNF-α) and glutamate in the rat model of haloperidol-induced tardive dyskinesia;	[128]
Psychiatric symptoms in COVID 19	Human studies:	
	The inappropriate function of the RAS may contribute to the exaggeration of psychiatric symptoms in patients with COVID-19	[129]
	The excessive stimulation of AT1R influences microglial polarization and induces an active M2a proinflammatory state and may thereby initiate neurodegenerative processes;	
Vasopressin system (VPS)		
Affective disorders	Human studies:	
	The increased expression of AVP mRNA in the PVN/SON in brains of patients with MDD;	[130]
	The association of *V1bR* gene polymorphism (haplotype associated with A-T-C-A-G for the single nucleotide polymorphism (SNP) s1-s2-s3-s4-s5 allele) with a protective effect for recurrent MDD;	[131]
	The association of the V1bR SNPs (rs28676508, rs35369693) with child aggression;	[132]
	The linkage of the V1bR genetic variation SNP rs33990840 with suicidal behavior;	[133]
	Elevated copeptin (surrogate marker of AVP) in patients resistant to antidepressant pharmacotherapy;	[134,135,136]
	V1bR antagonists are currently being trialed for the treatment of MDD;	[134,135]
	Rodent studies:	
	In rodent models of anxiety and depression, the antagonists of V1bR show anxiolytic- and antidepressant-like effects;	[137,138,139]
	The blockade of the central V1 receptors abolished anhedonia induced by chronic mild stress;	[140]
	The activation of brain VPS in stress;	[26,140,141,142,143,144,145]
	The blockade of V1bR induces anxiolytic actions in various models of depression;	[146,147]
Alzheimer’s disease	Human studies:	
	Low concentration of AVP in the CSF;	[148]
	The reduced expression of AVP immunoreactivity in the hippocampus, nucleus accumbens, and the internal portion of the globus pallidus of AD patients in comparison with controls (post-mortem studies);	[149]
	A reduced number of AVP expressing cells in the suprachiasmatic nucleus in senescence and AD patients;	[150]
	Vasopressinergic innervation of the PVN, SON, and locus coeruleus in AD patients and non-demented controls do not differ;	[151,152]
	Rodent studies:	
	The improvement of working memory and long-term memory in APP/PS1 mouse model of AD after the intranasal application of AVP-(4-8);	[153]
	The improvement of social memory is enhanced by the stimulation of V1bR in the hippocampus in mice;	[154]
Schizophrenia	Human studies:	
	Lower AVP levels in the temporal cortex of schizophrenic patients (post-mortem studies);	[155]
	Reduced AVP mRNA in the PVN of schizophrenic patients (post-mortem studies);	[156]
	In patients with schizophrenia, blood AVP levels are either elevated or not altered;	[157,158,159]
	A positive correlation between blood AVP level and severity of symptoms is found in female but not in male schizophrenia patients.	[159]
	Increased blood AVP levels, polydipsia, hypoosmolality, and hyponatremia are found in some patients with schizophrenia	[160,161,162]
	The intranasal application of DDAVP (synthetic analog of AVP) increases the effectiveness of risperidone in reducing the negative symptoms of schizophrenia;	[163]
	The associations between SNPs of the AVP gene and schizophrenia (chromosomal region 20p13, loci rs2740204 and rs3011589);	[164]
	Rodent studies:	
	Schizophrenia-like symptoms with impairment of social behavior in AVP-deficient (di/di) Brattleboro rats and *V1aR* knockout mice;	[165,166,167]
	The lower expression of AVP receptors in the prefrontal cortex and hypothalamus in the MAM model of schizophrenia in rats;	[168]
Autism spectrum disorder	Human studies:	
	Lower AVP concentrations in CSF of children with autism, and AVP levels were associated with the severity of symptoms;	[169,170,171]
	AVP concentration in the CSF in neonates predicts a subsequent diagnosis of autism;	[170]
	A significant association between ASD with polymorphism of the *V1aR* and *V1bR* genes and autism (SNP rs35369693 and rs28632197);	[172,173,174,175]
	Intranasally applied AVP improves social abilities and reduces anxiety symptoms in children with ASD;	[176,177]
Oxytocin system (OTS)		
Alzheimer’s disease	Human studies:	
	In AD patients, intranasally applied OT does not influence the activity of the brain regions affected by AD;	[151,178,179,180,181,182]
	In AD patients, magnetic resonance images show that the plasma OT concentration correlates with the right parahippocampal gyrus volume;	[183]
Affective disorders	Human studies:	
	The increased activity of the central OTS in depressive mood disorders;	[130,184,185,186]
	Inconsistent data on the correlation between plasma OT levels and depression;	[184,187,188,189,190,191,192]
	Plasma OT levels positively correlate with help-seeking intentions, behavior, and estimation of happiness in patients with depression or anxiety;	[184,193,194]
	The positive associations between depression, MDD, and separation anxiety and single nucleotide polymorphism (rs53576; rs2254298; rs53576 genotype A allele) of the OTR gene and with G-protein genes (Gβ3 rs5443);	[195,196,197,198,199,200]
	Low plasma OT levels in the third trimester of pregnancy may predict postpartum depressive symptoms;	[201]
	The level of blood oxytocin is lower in mothers with post-partum depression than in nondepressed mothers;	[202]
	Rodent studies:	
	Endogenous OTS decreases anxiety behavior in pregnant and lactating rats;	[201]
Schizophrenia	Human studies:	
	The association of schizophrenia with single nucleotide polymorphisms of the OT gene in chromosomal region 20p13 (rs4813626);	[164]
	Lower plasma OT concentrations in patients with schizophrenia;	[158,203]
	The negative correlation between OT levels and the severity of symptoms in patients with schizophrenia;	[157]
	The positive effects of the intranasally applied OT on social cognition in patients with schizophrenia;	[204,205]
	Clinical trials and meta-analyses do not support the significant therapeutic effect of OT in schizophrenia;	[206,207,208,209]
	Rodent studies:	
	Reduced concentrations of OT and OTRs in the prefrontal cortex and in the hypothalamus of rats in the experimental MAM model of schizophrenia;	[168]
Autism spectrum disorder	Human studies:	
	A strong association between OTR gene polymorphism (rs2254298, rs2268491, rs53576, rs237887, rs2268493, rs1042778 and rs7632287) and susceptibility to ASD;	[210,211,212,213,214,215,216,217]
	Lower plasma oxytocin levels in children with ASD;	[218,219,220,221]
	Intravenously infused or intranasally applied OT ameliorates repetitive behavior in adults with ASD and Asperger’s disorder;	[222,223]
	Randomized crossover trials show that in children with ASD, treatment with intranasal OT improves caregiver-rated social responsiveness and enhances learning in response to social targets and feedback;	[176,224]
	The lack of beneficial effects of OT in ASD;	[225,226,227]
	A systematic review and meta-analysis of tolerance of long-term intranasal application of OT in ASD.	[228]

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
