# Peer review of "Multiple Aspects of Inappropriate Action of Renin–Angiotensin, Vasopressin, and Oxytocin Systems in Neuropsychiatric and Neurodegenerative Diseases"

_jcm, 2022, doi:10.3390/jcm11040908_

Round 1

Reviewer 1 Report

Manuscript ID: jcm-1557327

In this review article, Ewa Szczepanska-Sadowska et al. presented a detailed review on the different regulatory/modulatory roles of the renin–angiotensin system (RAS), the vasopressin system (VPS), and the oxytocin system (OTS) systems in various neuropsychiatric conditions and neurodegenerative disorders. Authors performed an extensive literature review for this article and the manuscript is very well written. There are no significant or notable grammatical or typographical errors observed.  Overall, this is a well-crafted review article and I do recommend for publication of this article after addressing below listed concerns.  

Comments:

  1. Authors covered some important psychiatric conditions in this review along with Parkinson’s and Alzheimer’s disorders. Parkinson’s and Alzheimer’s disorders are considered as neurodegenerative (not neuropsychiatric) diseases. Hence, authors should modify the title to include neurodegenerative term or use a broader term which includes both ‘neurpsychiatric and neurodegenerative’ conditions.   

  1. Authors must consider including discussion on actions of RAS, VPS and OTS systems with other important psychiatric conditions such as Bipolar disorder, Major depressive disorder (although there is a brief discussion in the manuscript), ADHD, Seizures/Epilepsy, Anxiety and Migraine headaches to make it a comprehensive review.   

  1. Schizophrenia is one of the important neuropsychiatric disorders and authors presented only a brief discussion and review on this topic. Authors must elaborate.

  1. Although the information provided on coronavirus infections/coronavirus 2 (SARS-CoV-2) is important, it does not fall under neuropsychiatric conditions category and there is no relevance to this discussion and the title of the article.

  1. The information and literature review provide in the article is good, however authors should provide/organize RAS, VPS and OTS systems and their functional actions on different disease conditions in a tabulated form with suitable references for easy and clear understanding.

  1. Authors should consider including additional figures explaining mechanistic aspects of RAS, VPS and OTS systems in some important disease conditions.

  1. Line 376: Is it same or similar. Please correct it.

  1. Line 436: ‘bipolar affective disorder depressive episodes’, it seems few words are mixed up. Please correct them.

  1. Authors presented a discussion on the actions of Vasopressin and Oxytocin in ASD. Some studies show that RAS also plays key role in ASD and why authors did not include the discussion of RAS and ASD.              

Author Response

The authors are deeply obliged to the Reviewers for the comments to their study.

Reviewer #1

Comment 1: Authors covered some important psychiatric conditions in this review along with Parkinson’s and Alzheimer’s disorders. Parkinson’s and Alzheimer’s disorders are considered as neurodegenerative (not neuropsychiatric) diseases. Hence, authors should modify the title to include neurodegenerative term or use a broader term which includes both ‘neuropsychiatric and neurodegenerative’ conditions.   

Answer 1. We agree that the title does not fully correspond to the content of the Review and we made the following modification: “ Multiple Aspects of Inappropriate Action of Renin-Angiotensin, Vasopressin, and Oxytocin Systems in Neuropsychiatric and Neurodegenerative Diseases”. We inserted also corrections in the text of the manuscript (see Abstract, lines 21, 28, 31, and lines 33, 71, 78… 666 of the manuscript).   

Comment 2. Authors must consider including discussion on actions of RAS, VPS and OTS systems with other important psychiatric conditions such as Bipolar disorder, Major depressive disorder (although there is a brief discussion in the manuscript), ADHD, Seizures/Epilepsy, Anxiety and Migraine headaches to make it a comprehensive review.   

Answer 2. The authors realize that their Review does not discusses sufficiently enough all essential neurological, neuropsychiatric and neurodegenerative processes, in which functions of  RAS, VAS and OXY are altered. The search of literature showed us that the topic is very abounding and complex  and we decided to concentrate in this Review on the diseases for which the information is most convincing.  We were also afraid that discussion of too many pathological processes in one work would make the study not enough transparent. We are going to prepare another study, in which we could approach the problems of bipolar disorder, ADHD, seizure /epilepsy, anxiety and migraine headaches.

Comment 3. Schizophrenia is one of the important neuropsychiatric disorders and authors presented only a brief discussion and review on this topic. Authors must elaborate.

Answer 3. Schizophrenia belongs to difficult neuropsychiatric diseases which affect human populations of different age and sex. Unfortunately, relative little attention has been given to the regulation of the cardiovascular system in the schizophrenic patients and in spite of intense search of the literature we could not find many studies devoted to the role of RAS in schizophrenia in context of cardiovascular regulation. The authors focused mainly on behavioral and social cognition symptoms (see for instance: Winship et al., Can J Physiol 2019, 64:5-17; Nestler and Hyman Nat Neurosci. 2010;13(10):1161-9; Kumari V and Ettinger, Schizophrenia Res  2020; 221. 4-11). Studies devoted to the role of RAS, VPS and OTS are not univocal and speculative. In the revised version we intended to emphasize the lack of this information (2.4.6. “RAS in schizophrenia and autism”, lines 327-329, 335-336, 345-348, and 3.2.5.”Vasopressin and oxytocin in schizophrenia”). We introduced also the reference to the study of Oh and Fan (2019) and a new information about association of psychogenic polydipsia with RAS in schizophrenic patients  (section 2.4.6, 161, 167).

The Possible Role of the Angiotensin System in the Pathophysiology of Schizophrenia: Implications for Pharmacotherapy. Oh SJ, Fan X.CNS Drugs. 2019 Jun;33(6):539-547. doi: 10.1007/s40263-019-00632-4.

Treatment of psychogenic polydipsia: comparison of risperidone and olanzapine, and the effects of an adjunctive angiotensin-II receptor blocking drug (irbesartan). Kruse D, Pantelis C, Rudd R, Quek J, Herbert P, McKinley M.Aust N Z J Psychiatry. 2001 Feb;35(1):65-8. doi: 10.1046/j.1440-1614.2001.00847.x

Comment 4. Although the information provided on coronavirus infections/coronavirus 2 (SARS-CoV-2) is important, it does not fall under neuropsychiatric conditions category and there is no relevance to this discussion and the title of the article.

Answer 4. We think that presentation of research on the role of RAS in coronavirus disease goes well together with other problems discussed in our review because pathology of SARS-CoV-2 infection can affect the patients with neuropsychiatric and neurodegenerative diseases and involves dysfunction of RAS. Thus, it is likely that inappropriate activation of  RAS may account for some disorders of the brain function observed in COVID-19. To more expose associations of RAS with COVID-19 we made the following rearrangements in the text:

  • We introduced COVID-19 into the Abstract (line 23).
  • We introduced the sentence “Growing evidence indicates that inappropriate activation of RAS components may contribute to disturbances of the brain functions in coronavirus disease (COVID-19) (lines 355-356).
  • We removed the sentence “Neurons and glial cells are among the target cells of severe acute respiratory syndrome associated coronavirus 2 (SARS-CoV-2)”.
  • We altered the next sentence in the following way: “Survivors of COVID-19 manifest aggravated symptoms of neuropsychiatric disorders such as cognitive and attention deficits (i.e., brain fog), new-onset anxiety, depression, psychosis, seizures, and suicidal behavior [169].”
  • We removed the next sentence “Neuropsychiatric symptoms related to COVID-19 infection are observed before, during, and after respiratory symptoms and are unrelated to respiratory insufficiency, suggesting independent brain injuries.”
  • At the end of paragraph 2.4.7. we introduced the sentence “ Altogether, it appears that inappropriate activity of RAS should be taken into account as an effective cause of the brain dysfunction occurring in COVID-19 disease.”  

Comment 5. Although the information and literature review provide in the article is good, however authors should provide/organize RAS, VPS and OTS systems and their functional actions on different disease conditions in a tabulated form with suitable references for easy and clear understanding.

Answer 5. We introduced the Table with functional actions of RAS, VPS and OTS, as it was recommended by the Reviewer.

Comment 6. Authors should consider including additional figures explaining mechanistic aspects of RAS, VPS and OTS systems in some important disease conditions.

Answer 6.  The most important information addressing mechanistic aspects of RAS, VPS and OTS are included in Figure 2. In the revised version of the Legend to this Figure we refer to our previous studies, in which the mechanistic aspects of action of RAS, VPS and OTS in cardiovascular diseases have been illustrated (line 117 ref. 16-17, 190).

Comment 7. Line 376: Is it same or similar. Please correct it.

Answer 7. “The same” is now replaced by “similar”. (line 402)

Comment 8. Line 436: ‘bipolar affective disorder depressive episodes’, it seems few words are mixed up. Please correct them.

Answer 8. We altered sequence of words in this sentence in the following way: “Most of the studies report lower levels of plasma OT in patients with MDD, bipolar affective disorder and depressive episodes than in the control subjects [234–237], however higher [238–239] or normal [240] levels were also found”

Comment 9. Authors presented a discussion on the actions of Vasopressin and Oxytocin in ASD. Some studies show that RAS also plays key role in ASD and why authors did not include the discussion of RAS and ASD.

Answer 9. The literature concerning the role of ACE, VPS and OTS in autism is scanty. We could find only one publication in which genetic variants of ACE were described and it is now introduced into the revised version as the position  168

Genetic Variants of Angiotensin-Converting Enzyme Are Linked to Autism: A Case-Control Study. Firouzabadi N, Ghazanfari N, Alavi Shoushtari A, Erfani N, Fathi F, Bazrafkan M, Bahramali E.PLoS One. 2016 Apr 15;11(4):e0153667. doi: 10.1371/journal.pone.0153667. eCollection 2016.

Reviewer 2 Report

The authors did a very good job revising and summarizing the extensive literature available for these topics. I like who they relate the role of angiotensin, vasopresin and oxcitosin with brain and neuropsychiatric disorders. And I appreciate that they make a great effort to include different sources of information, i.e.  pharmacogenetic results, animal models and expression studies. I only have few minor suggestions:

I would suggest to reduce the number of abbreviations used, since in some points it can be difficult for the person who is reading the review to remember what does they refer. Just as an example, maybe is not necessary to abbreviate major depression as MDD, it would be easier to follow for the reader. Also you can remove those that only appears once like: knockout (KO).

In my opinion, I would remove the section named "2.4.7. RAS in coronavirus infections" since it is not directly related with the scope of the review, neuropsychiatric disorders, and everything included there seems still some kind of speculations.

Gene names should be written in italics, for example, when refering to a knock-out mice:  Avpr1b -/- mice.

Author Response

The authors are deeply obliged to the Reviewers for the comments to their study.

Reviewer # 2

The authors did a very good job revising and summarizing the extensive literature available for these topics. I like who they relate the role of angiotensin, vasopressin and oxytocin with brain and neuropsychiatric disorders. And I appreciate that they make a great effort to include different sources of information, i.e.  pharmacogenetic results, animal models and expression studies. I only have few minor suggestions:

Comment 1: I would suggest to reduce the number of abbreviations used, since in some points it can be difficult for the person who is reading the review to remember what does they refer. Just as an example, maybe is not necessary to abbreviate major depression as MDD, it would be easier to follow for the reader. Also you can remove those that only appears once like: knockout (KO).

Answer 1. In our opinion introduction of MDD abbreviation of major depression should be left in text. However according to Reviewers suggestion to reduce the number of abbreviation we have removed abbreviations that appear once. 

Comment 2: In my opinion, I would remove the section named "2.4.7. RAS in coronavirus infections" since it is not directly related with the scope of the review, neuropsychiatric disorders, and everything included there seems still some kind of speculations.

Answer 4. We think that presentation of research on the role of RAS in coronavirus disease goes well together with other problems discussed in our review because pathology of SARS-CoV-2 infection can affect the patients with neuropsychiatric and neurodegenerative diseases and involves dysfunction of RAS. Thus, it is likely that inappropriate activation of  RAS may account for some disorders of the brain function observed in COVID-19. To more expose associations of RAS with COVID-19 we made the following rearrangements in the text:

  • We introduced COVID-19 into the Abstract (line 23).
  • We introduced the sentence “Growing evidence indicates that inappropriate activation of RAS components may contribute to disturbances of the brain functions in coronavirus disease (COVID-19) (lines 355-356).
  • We removed the sentence “Neurons and glial cells are among the target cells of severe acute respiratory syndrome associated coronavirus 2 (SARS-CoV-2)”.
  • We altered the next sentence in the following way: “Survivors of COVID-19 manifest aggravated symptoms of neuropsychiatric disorders such as cognitive and attention deficits (i.e., brain fog), new-onset anxiety, depression, psychosis, seizures, and suicidal behavior [169].”
  • We removed the next sentence “Neuropsychiatric symptoms related to COVID-19 infection are observed before, during, and after respiratory symptoms and are unrelated to respiratory insufficiency, suggesting independent brain injuries.”
  • At the end of paragraph 2.4.7. we introduced the sentence “ Altogether, it appears that inappropriate activity of RAS should be taken into account as an effective cause of the brain dysfunction occurring in COVID-19 disease.” 

Comment 3. Gene names should be written in italics, for example, when referring to a knock-out mice:  Avpr1b -/- mice.

Answer 3. Changes have been made.

Reviewer 3 Report

The present review presented evidence that there are multiple causes of altered function of the RAS, VPS, and OTS in psychiatric and neurodegenerative disorders.

I think the manuscript includes new and intriguing contents and that is very informative for researchers as well as clinicians., however the authors should revise it according to the following concerns;

  1. The authors should cite the following literatures which describe a role of oxytocin in neuropsychiatric symptoms.

 PMID:32233820

  1. The authors should discuss more in detail on the interaction of RAS and VPS/OTS in the pathology of psychiatric and neurodegenerative disorders, citing relevant literatures.

Author Response

The authors are deeply obliged to the Reviewers for the comments to their study.

Reviewer # 3

Comment 1. The authors should cite the following literatures which describe a role of oxytocin in neuropsychiatric symptoms.

 PMID:32233820

Answer 1. We have introduced the recommended study in the text (line 418) and in the References (ref. 210)

Serum Oxytocin Levels and Logical Memory in Older People in Rural Japan. Kunitake Y, Imamura Y, Mizoguchi Y, Matsushima J, Tateishi H, Murakawa-Hirachi T, Nabeta H, Kawashima T, Kojima N, Yamada S, Monji A.J Geriatr Psychiatry Neurol. 2021 Mar;34(2):156-161. doi: 10.1177/0891988720915526.

Comment 2. The authors should discuss more in detail on the interaction of RAS and VPS/OTS in the pathology of psychiatric and neurodegenerative disorders, citing relevant literatures.

Answer 2. We have introduced some new contents into the text of the study which should enrich its value. They are included into paragraphs 2.4.6 (lines 326-351), 2.4.7,  3.1 (line 418), and 4.0 (lines 660-663) and Table 1.